# Disparity in school children's reading skills in 11 African countries

**Huafeng Zhang**[1,2]*, **Stein T. Holden**[1]

**1** School of Economics and Business, Norwegian University of Life Sciences, Ås, Norway, **2** Fafo Institute for Labour and Social Research, Tøyen. Oslo, Norway

* zhu@fafo.no

## Abstract

To promote SDG Goal 4 and "education for all", this study investigates children's basic reading skills in 11 low-income and lower-middle-income African countries, using standardized reading tests from the Multiple Indicator Cluster Surveys (MICS). Research specifically examining children's reading skills and disparities across socioeconomic groups in African contexts remains scarce. This study addresses a critical knowledge gap by providing comparative evidence on reading skills disparities across diverse social backgrounds, including children with disabilities. Our study provides new evidence on the "Learning Crisis in the Global South", revealing alarmingly low levels of reading skills but with considerable variation across the 11 African countries studied. Substantial reading skills differences exist between children with disabilities or from disadvantaged backgrounds—those living in rural areas, and from poorer, less educated families—and their non-disabled and non-disadvantaged peers. Notably, these disparities are often more pronounced in countries with higher overall reading proficiency. Moreover, there are persistent gaps between children with and without disabilities across the countries and socioeconomic groups in this study. Encouragingly, strengthening education systems is a promising way of improving the reading skills of children with disabilities. These findings underscore the diverse challenges faced by children from different backgrounds in varying contexts.

## 1. Introduction

The UN Sustainable Development Goal 4 underscores the importance of achieving inclusive and equitable quality education for all [1,2]. There is a growing interest in understanding the educational outcomes of children from disadvantaged backgrounds and identifying the factors contributing to variations in these outcomes, which can inform the development of effective educational policies [3,4,5]. In recent years, following the debate on the "Learning Crisis in the Global South" [6,7], reading proficiency has emerged as a crucial focus in sub-Saharan Africa, recognised as a key indicator of learning outcomes and the success of formal education. The percentage of students attaining the minimum proficiency level in reading skills is a key indicator for achieving SDG Goal 4, given the emphasis on reading skills by the UNESCO Global Education Monitoring Report (2014) [7].

**Funding:** The paper has been undertaken as part of the research project "Education outcome variability in children with disabilities: Structure, institution or agency?" funded by the Research Council of Norway (Project Number 300635). The funders had no role in study design, data collection and analysis, decision to publish, or preparation of the manuscript.

**Competing interests:** The authors have declared that no competing interests exist.

Previous research in developed contexts has emphasised the persistent differences in reading skills between children from disadvantaged and non-disadvantaged backgrounds [8,9,10]. In developing countries, efforts have traditionally centred on socioeconomic factors such as gender, education, income, and geographical location [11,12,13]. Numerous cross-country studies on children's reading performance have offered valuable insights into the role of gender, home environment, school socioeconomic status, and literacy interventions in shaping children's reading [14–19]. However, these studies often rely on international standard learning assessments, such as PIRLS (the Progress in International Reading Literacy Study) and PISA (Programme for International Student Assessment). These assessments primarily target developed or OECD countries, with limited participation from African nations. Of the 102 countries that have ever participated in PISA, only eight are from Africa, including just four from Sub-Saharan Africa. PIRLS has even fewer African participants.

Due to data constraints, research specifically examining children's learning performance, such as reading or numeracy skills, and the disparities in these outcomes across socioeconomic groups in African contexts remain limited. Some studies rely on data from the Confemen Programme for the Analysis of Educational Systems (PASEC), which surveyed 10 and 14 African countries in 2014 and 2019, respectively. Using PASEC, reading skills were reported as significantly lower among children from poor or disadvantaged families [20,21]. Furthermore, Kadio (2023) highlighted that gender disparities in educational outcomes are correlated with socioeconomic status, with children from disadvantaged backgrounds experiencing disproportionally higher gender-based disparities [22]. The challenges faced by children with disabilities (CWD) and their low learning performance have only recently garnered attention, particularly following the adoption of the United Nations Convention on the Rights of Persons with Disabilities (UNCRPD) in 2006 [23]. Recent studies have made efforts to understand the schooling challenges faced by CWD, focusing on differences in school access, attendance and enrolment in developing countries [23–25]. However, studies specifically addressing how much CWDs are falling behind in reading skills learning are rare in the context of developing countries, with a few from individual studies in Asia [26].

While none of these PASEC-based studies explicitly focused on children with disabilities, Wodon et al. (2018) reported large disparities in reading and numeracy learning between children with hearing or vision disabilities and their peers without disabilities in 10 African countries [27]. Other cross-country comparative studies have utilized MICS survey data, as in the present study. UNICEF (2022) reported different disparities in reading and numeracy skills across different disability types [28]. However, the report primarily provided global estimates or findings from a limited number of countries, without a specific focus on African countries. Zhang and Holden (2023), also using MICS data, found that barriers to numeracy skills among children with disabilities vary by disability type: some children are hindered primarily by lack of school access; while others face dual barriers related to school access and skill acquisition within schools [29].

Using nationally representative data across 11 low-income and lower-middle-income African countries, we evaluate the reading skills of children aged 10–14 years old and investigate variations in reading skills across rural versus urban areas, between children with disabilities (CWD) versus children without disabilities (CWOD), as well as between children from poorer and less educated families versus better-off and more educated families. More specifically, this study provides unique insights into how these disparities differ across 11 African countries and highlights the relative performance of CWD compared to CWOD within various social groups and across different national contexts.

Our research aims to answer the following research questions: 1) To what extent do children with disabilities or from disadvantaged backgrounds (e.g., children from poorer or less educated families, rural areas) lag behind their peers (children without disabilities or from better off or educated families, urban areas) in acquiring basic reading skills? 2) Do children with disabilities or from disadvantaged backgrounds benefit equally from higher national-level reading proficiency? 3) Can better micro-level social conditions help mitigate the learning constraints faced by children with disabilities?

This paper is unique in its exclusive focus on school children's reading skills performance across low-income and lower-middle-income African countries, all of which were included in the sixth round of Multiple Indicator Cluster Surveys (MICS) between 2017 and 2020. First, we present comprehensive, nationally representative evidence of the substantial variation in basic reading skills among children from different socioeconomic backgrounds. We employ consistent, standardised tests and measurements of reading skills both within and across countries. We identify substantial differences in reading skills across the 11 countries, as well as across socioeconomic groups within each country.

Second, we utilise the standardised identification of children with disabilities in the MICS survey to assess their reading skills, using children without disabilities in each country as a counterfactual. Overall, children with disabilities lag behind children without disabilities. However, an interesting finding is that children with disabilities in better-performing countries outperform children without disabilities in other countries. This suggests that strengthening education systems is a promising way of improving the reading skills of children with disabilities.

## 2. Conceptual framework

Reading skills are crucial for the development of various other academic skills in school and can greatly impact children's likelihood of repeating grades or dropping out [30]. Several social, familial and individual factors influence children's learning, and the mechanisms through which these factors influence learning are multifaceted (Taylor [Unpublished]). Pace et al. (2017) identify three potential pathways by which socioeconomic status might influence children's language development, which are child characteristics, parent-child interaction, and the availability of learning resources [31].

This paper aims to evaluate children's reading skills performance in any of the three potential pathways as suggested by Pace et al. (2017). First, children who have functional challenges in one of the four main functional domains – vision, hearing, physical, intellectual – or with multiple functional challenges. Second, children from families in the lowest quintile of the asset index, and children from families without schooling. These children quite often have little access to critical learning resources and parental engagement for language development. Finally, children living in rural areas, where learning resources are constrained and school quality is often lower.

Families with higher social status, including better income and higher education levels, tend to provide better support for their children's learning. Children from more advantaged backgrounds often begin their learning process earlier than their peers from disadvantaged families [32]. Additionally, they may indirectly benefit from residing in neighbourhoods with higher-quality schools [33]. Parents with higher social status are also more likely to actively engage with the school community, thereby contributing to overall school quality.

The neighbourhood environment can influence children's learning outcomes. In the African context, although not extensively studied, there is evidence of urban-rural disparities in schooling [11]. Rural areas often face challenges related to school quality due to a lack of infrastructure, educational resources, and qualified teachers. Furthermore, in neighbourhoods

characterised by high levels of poverty in rural areas, various social issues affecting disadvantaged families can be exacerbated. Children are also exposed to the influences of their peers in the same neighbourhood or school [34].

The challenges related to learning reading skills vary greatly across different disability types due to the diverse nature of functional difficulties [35,36]. Children with vision disabilities may have the same capability to develop reading skills as their peers, but the real challenges often stem from the availability of aids, such as corrective lenses, optical devices, and glasses [37], as well as access to consultative instructional services [38]. For children with hearing disabilities, the challenge of learning to read often arises from a lack of exposure to their first language before the critical period [39]. This puts them at high risk of linguistic deprivation [40]. Children with physical disabilities may not face apparent functional challenges in learning reading skills, but they frequently experience high rates of school absenteeism due to factors like long distances to school and lack of infrastructure, materials, and support [41]. Children with intellectual disabilities struggle with developing reading skills due to challenges in various abilities, including information processing, cognitive abilities, and attentive behaviours [42,43]. Children with multiple disabilities are exposed to higher risks related to several different functional challenges. Moreover, the availability of appropriate teaching materials and pedagogical interventions for CWD can enhance their skill development.

We set up the first hypothesis concerning the role of factors related to child characteristics, parent-child interaction, and the availability of learning resources:

*H1. The percentages of school children aged 10–14 with satisfactory reading skills among children with a) families in the lowest quintile of the asset index, b) families without schooling, c) rural residence, d) disabilities (vision, hearing, physical, intellectual, and multiple disabilities) are significantly lower than that among their peers.*

Several cross-country studies focusing on school enrolment have shown that disparities in enrolment and attendance for disadvantaged children are more pronounced in countries with higher overall enrolment rates and better socio-economic development [23,44,45]. We formulate the second hypothesis to explore whether children from different backgrounds benefit equally from their country's overall reading proficiency level:

*H2. The differences in the percentage of school children with satisfactory basic reading skills are more pronounced in countries with higher overall reading proficiency when comparing a) children from families in the lowest asset quintile vs. those in the upper quintiles, b) children from families without vs. with schooling, c) rural vs. urban children, and d) CWD vs. CWOD.*

Another question revolves around whether CWD, when raised in families with a more advantageous social background (urban residence, higher income, higher education), can successfully bridge the academic performance gap compared to CWOD. Can better micro-level socioeconomic conditions help mitigate the learning constraints faced by children with disabilities? We set up the third hypothesis related to the reading skills associated with children's disabilities across different social groups:

*H3. The differences in the percentage of school children with satisfactory basic reading skills between CWD and CWOD are smaller in a) urban, b) higher-income, c) more educated families.*

Our H3a-c hypotheses are based on the notion that families with advantageous conditions can better support CWD in overcoming learning challenges. Finally, due to data limitations, our assessment is confined to children enrolled in school during the survey period.

## 3. Materials and methods

### 3.1 Data description

We use publicly available data from the sixth round of MICS national representative surveys conducted by the United Nations International Children's Emergency Fund (UNICEF) between 2017 and 2020 in 11 African countries: Central Africa Republic, Chad, DRCongo, Ghana, Lesotho, Madagascar, Malawi, The Gambia, Togo, Tunisia, Zimbabwe. The sixth round of MICS adopted the Washington Group Child Functioning Module (WG-CFM) to assess functional difficulties among children aged 6–17 [46,47]. Out of the 13 functional domains covered by WG-CFM, this paper focuses on eight domains that include four severity scales, categorised into five types of disabilities: vision, hearing, walking, intellectual and multiple disabilities [48].

Our analysis primarily relies on the reading test designed for children aged 10–14 in the MICS survey. This reading test is highly standardised and consistently applied across countries. It consists of an oral reading fluency test, where children read a short story of approximately 60–80 words [49], followed by a comprehension test containing five questions related to the story's content. From this test, we derive two key indicators: Q1, representing the proportion of correctly read words (ranging from 0 to 1), and Q2, indicating the proportion of correctly answered questions (with values of 0, 0.2, 0.4, 0.6, 0.8, 1). The reading test score is subsequently computed as the average of Q1 and Q2.

The distribution of these test scores shows a substantial number of extreme values, with children either reading fluently and answering all comprehension test questions correctly or being unable to read at all. The reading test in the MICS survey assesses foundational skills, and given the sample age of 10–14, after several years of schooling, all children should theoretically reach this basic level of reading. Children who struggle to achieve satisfactory proficiency in these tests face notable challenges in reading. Rather than treating the reading test score as a continuous measure, this study focuses on identifying children who are struggling with reading. We use the percentage of school children who surpass the threshold score of 0.85 as the primary outcome variable. Additionally, we include sensitivity analysis (reported in Table S2.3, S2.5, and S2.7 in S1 File) using continuous outcome variables. These results are consistent with the findings based on the threshold-based outcome measure.

Furthermore, although the 0.85 threshold is somewhat arbitrary, it allows a maximum of one incorrect comprehension question and a limited number of errors in reading the story (up to 10 percent of words). However, the threshold at 0.9 might be a little bit too strict because the child will have to read all the words 100% correctly if one question is wrong, or the child has to answer all 5 questions correctly. To ensure robustness, we conduct sensitivity analyses using alternative cutoff points (0.8, 0.9) to assess whether they would significantly change our primary findings. The results of these sensitivity analyses are detailed in Supporting Information Table S2.1, S2.2, S2.4, and S2.6 in S1 File. The sensitivity test shows no large sensitivity to the selection of different cutoff thresholds.

In the MICS survey, one child aged between 6 and 17 is selected from the participating households to take the reading test. Table 1 provides an overview of the total sample size by country and the size of non-response.

In many countries, the majority of children who have never attended school (99.6 percent) or have dropped out (78.5 percent) did not take the reading test, accounting for 16.0 percent of the sample. Additionally, 2.7 percent of children did not take the reading test because the test was not available in their primary teaching language. In most countries, the test is administered in an official foreign language, such as English or French [50]. Finally, 13.6 percent of non-responses were due to refusals, with 4.7 percent attributed to families refusing to involve their child, and 8.9 percent to children themselves refused to take the reading test.

The sample size of the children who completed the reading tests is presented in Table 2, categorized by urban vs. rural location, CWD vs. CWOD, children from the lowest asset quintile vs. those in the upper quintiles), as well as children from families with vs. without schooling, across the 11 African countries.

## 3.2 Ethics methods

MICS surveys data were publicly available online data base, with all the surveys conducted by UNICEF. These surveys underwent review and received approvals from ethics committees in each respective country. Furthermore, participants in these surveys were provided with information about the surveys and informed consent process was conducted during all the MICS

**Table 1. Sample size and non-response by countries.**

| Country | Missing due to Out of school1 | | Missing due to Language | | Missing due to refusal2 | | Done reading test | | Total |
|---|---|---|---|---|---|---|---|---|---|
| | Number | Percent (%) | Number | Percent (%) | Number | Percent (%) | Number | Percent (%) | |
| Central African Repub | 361 | 17.8 | 145 | 7.1 | 444 | 21.9 | 1081 | 53.2 | **2,031** |
| Chad | 2,568 | 54.1 | 107 | 2.3 | 490 | 10.3 | 1582 | 33.3 | **4,747** |
| DRCongo | 769 | 16.6 | 305 | 6.6 | 754 | 16.2 | 2813 | 60.6 | **4,641** |
| Ghana | 176 | 5.0 | 112 | 3.2 | 267 | 7.6 | 2937 | 84.1 | **3,492** |
| Lesotho | 42 | 2.2 | 0 | – | 287 | 14.9 | 1598 | 82.9 | **1,927** |
| Madagascar | 958 | 22.3 | 1 | 0.0 | 656 | 15.3 | 2686 | 62.5 | **4,301** |
| Malawi | 204 | 3.0 | 69 | 1.0 | 1498 | 22.4 | 4930 | 73.6 | **6,701** |
| The Gambia | 366 | 18.7 | 190 | 9.7 | 179 | 9.2 | 1220 | 62.4 | **1,955** |
| Togo | 119 | 6.6 | 5 | 0.3 | 110 | 6.1 | 1576 | 87.1 | **1,810** |
| Tunisia | 20 | 1.1 | 0 | – | 77 | 4.4 | 1651 | 94.5 | **1,748** |
| Zimbabwe | 137 | 5.6 | 43 | 1.8 | 105 | 4.3 | 2156 | 88.3 | **2,441** |
| Total | **5,720** | **16.0** | **977** | **2.7** | **4,867** | **13.6** | **24,230** | **67.7** | **35,794** |

Note1 including children never-in-school and dropouts.

2 including family and child refusal.

**Table 2. Number of tested children by location, disability status, socioeconomic factors and country, ages 10-14.**

| Country | Location | | Disability Status | | Poverty Status | | Family Schooling | |
|---|---|---|---|---|---|---|---|---|
| | Rural | Urban | CWD | CWOD | Lowest quintile | Upper quintiles | No school | Other |
| Central African Repub | 472 | 609 | 66 | 1,015 | 108 | 973 | 190 | 888 |
| Chad | 994 | 588 | 41 | 1,541 | 163 | 1,419 | 662 | 918 |
| DRCongo | 1,673 | 1,140 | 45 | 2,768 | 625 | 2,188 | 335 | 2,477 |
| Ghana | 1,502 | 1,435 | 219 | 2,718 | 664 | 2,273 | 912 | 2,018 |
| Lesotho | 1,142 | 456 | 39 | 1,559 | 421 | 1,177 | 212 | 1,377 |
| Madagascar | 1,871 | 815 | 138 | 2,548 | 372 | 2,314 | 505 | 2,166 |
| Malawi | 4,124 | 806 | 153 | 4,777 | 697 | 4,233 | 654 | 4,256 |
| The Gambia | 582 | 638 | 21 | 1,199 | 386 | 834 | 761 | 454 |
| Togo | 1,031 | 545 | 83 | 1,493 | 340 | 1,236 | 532 | 1,031 |
| Tunisia | 514 | 1,137 | 49 | 1,602 | 326 | 1,325 | 197 | 1,448 |
| Zimbabwe | 1,518 | 638 | 90 | 2,066 | 428 | 1,728 | 131 | 2,024 |
| Total | 15,423 | 8,807 | 944 | 23,286 | 4,530 | 19,700 | 5,091 | 19,057 |

"Lowest quintile" refers to children from families in the lowest quintile of the asset index, while "Upper quintiles" includes all children not in the lowest quintile.

"No school" refers to children from families without any schooling, while "Other" includes all children from families with some level of formal education.

surveys, following the MICS protection protocol. Detailed information is provided in section 2.4 in the survey report for each country and publicly available on MICS website.

### 3.3  Estimation strategy

The MICS data is a national sample of children aged 6–17. However, the non-response rate in the MICS reading tests is as high as 32 percent. The majority of out-of-school children and all children taught in minority languages are excluded from the reading tests. As a result, our analysis can only confidently speak about in-school children taught in the main language.

We are able to address one of the selection problems in the data, non-participation due to refusal. To address this potential selection issue due to refusals, we employ inverse probability weighting (IPW). IPW relies on estimating the probability of exposure (in this case, taking the reading test) for each person in the sample by using probit regression models.

We first use a probit model to evaluate the likelihood of children in the sample taking the reading test in each respective country in the following:

$$\begin{aligned} Selection_i^m = \alpha_0^m + \alpha_{1j}^m D_{ij}^m + \alpha_3^m UR_i^m + \alpha_{2k}^m EDU_i^m \\ + \alpha_{2k}^m ASS_i^m + \alpha_4^m Age_i^m + \alpha_5^m Gender_i^m + \varepsilon_i \end{aligned} \tag{1}$$

To address potential sample selection, we include variables that could be correlated with a child's probability of taking the reading tests. Several factors may have contributed to children's participation rates in the reading tests. Children with lower reading abilities may have felt reluctant or ashamed to participate, potentially due to fear of embarrassment or negative judgment. To address this concern, we account for key factors that are commonly associated with children's reading skills in the selection model, including age, gender, and the family's socioeconomic status [10,51].

Additionally, children may have missed the test due to health-related issues or because they were engaged in household duties or other work responsibilities, factors that are particularly prevalent in low-resource settings. To account for these dynamics, we include children's disability status, urban or rural residence, and socioeconomic indicators, as these variables are often linked to the likelihood of children participating in domestic or economic labour [52,53].

Therefore, the control variables in the selection model encompass: 1) asset index indicator quintiles ( $ASS_i$ ), constructed using weighted assets owned by the household through the first principal component based on principal component analysis (PCA) at the household level [54]; 2) the highest completed educational level among the household members ( $EDU_i$ ); 3) location variable $UR_i$ , indicating urban or rural residence; 4) disability status ( $D_{ij}$ ), represented by dummy variables indicating no disability, vision, hearing, physical, intellectual, or multiple disabilities; and 5) children's age and gender. Here, subscript $i$ represents each individual child, m represents countries, $j$ represents different disability statuses.

If the coefficients for these variables are statistically significant, it indicates evidence of sample selection. The predicted probability of selection from the full model (1) is $\widehat{Selection1_i^m}$ . Next, we rerun a reduced probit model with covariates that are insignificant in (1) and the predicted probability from the reduced model is $\widehat{Selection2_i^m}$ . The inverse probability weight is calculated as the ratio between the two predicted probabilities:

$$Weight_i^m = \widehat{Selection2_i^m} / \widehat{Selection1_i^m} \tag{2}$$

The inverse probability weight is used on the sample consisting of children who have completed the reading test. The approach helps adjust for potential selection bias related to family

and individual characteristics since children with similar characteristics to those who refused the reading test will receive higher weights [55].

In the second stage model, only school children with reading test scores will be included, weighted by IPW.

We first test hypothesis H1, which states that the percentages of school children aged 10–14 with satisfactory reading skills among children with a) families in the lowest quintile of the asset index, b) families without schooling, c) rural residence, d) disabilities (vision, hearing, physical, intellectual, and multiple disabilities) are significantly lower than that among their peers.

We employ country-fixed effects models and include Asset index quintile ($ASS_i$), Families' educational level ($EDU_i$), urban/ rural residence ($UR_i$), disability status ($D_{ij}$), as well as additional control variables such as age and gender in the models. Initially, we run four separate models, each including only one of these factors alongside the control variables, to test the treatment effect of each factor individually. Then, we run the model with all factors and control variables included, using the following model specification:

$$Reading_i = \beta_0 + \beta_{1j}D_{ij} + \beta_2 UR_i + \beta_3 ASS_i + \beta_4 EDU_i \\ + \beta_5 Age_i + \beta_6 Gender_i + \beta_7 Country_i + u_i \tag{3}$$

Here, subscript $i$ represents each individual child.

To test hypothesis H2, which states that the differences in the percentage of school children with satisfactory reading skills between disabled and non-disabled as well as between disadvantaged and non-disadvantaged backgrounds are more pronounced in countries with higher overall reading proficiency, we include interaction terms between different factors and country variable. Similarly, we run four separate models, each including the interaction term between the country and one factor $F_{ij}$ (j represents one of the factors: poverty status, family schooling, urban/ rural residence, and disability status). The model specification is as follows:

$$Reading_i = \pi_{10} + \pi_{11}F_{ij} + \pi_{12}Country_i + \pi_{13}F_{ij} * Country_i + \pi_{14}UR_i \\ + \pi_{15}ASS_i + \pi_{16}EDU_i + \pi_{17}Age_i + \pi_{18}Gender_i + u_{1i} \tag{4}$$

The sample size is relatively small for some groups in certain countries, particularly for children with disabilities, resulting in a high variance in the estimations. Therefore, we also categorise the 11 countries in the sample into three country groups (CGrp): low-reading country, mid-reading country, and high-reading country. We run separate models again, similar to (4) with the country group variable. The new set of regressions follows the model specification:

$$Reading_i = \pi_{10} + \pi_{11}F_{ij} + \pi_{12}CGrp_i + \pi_{13}F_{ij} * CGrp_i \\ + \pi_{14}UR_i + \pi_{15}ASS_i + \pi_{16}EDU_i + \pi_{17}Age_i + \pi_{18}Gender_i + u_{1i} \tag{5}$$

To test hypothesis H3, which states that the differences in the percentage of school children with satisfactory reading skills between CWD and CWOD are smaller in a) urban, b) higher-income, c) more educated families, we include interaction terms between disability status and other micro-level indicators:

$$Reading_i = \pi_{20} + \pi_{21}D_i + \pi_{22}UR_i + \pi_{23}ASS_i + \pi_{24}EDU_i + \pi_{25}D_i * UR_i \\ + \pi_{26}D_i * ASS_i + \pi_{27}D_i * EDU_i + \pi_{28}Age_i + \pi_{29}Gender_i + \pi_{30}Country_i + u_{2i} \tag{6}$$

Due to the limitations in the size of samples for some disability types, we will not estimate the treatment effect of different disability types but include disability status $D_i$ as a catch-all category.

## 4. Results

### 4.1 Reading skills across 11 African countries

The percentage of school children aged 10–14 with satisfactory reading skills (reading score 0.85 or above) in each country is displayed in Table 3, showing substantial variation. This percentage ranges from a low of 17.8% in the Central African Republic to a high of 87.7% in Tunisia. Seven countries have more than 50% of children with unsatisfactory reading skills. In our combined sample from 11 countries, fewer than half (45 per cent) of school children have achieved a satisfactory reading level. Namely, they can read the basic text properly. As shown in Table 3, the reading proficiency levels among school children highlight not only the generally low overall reading skills but also substantial variations across the 11 African countries.

Based on the overall reading skills proficiency of these countries, we can categorize them into three groups: low-reading countries, which include the Central Africa Republic, Chad, DRCongo, and The Gambia; mid-reading countries, which include Ghana, Madagascar, Malawi, and Togo; and high-reading countries, which include Lesotho, Tunisia, and Zimbabwe.

### 4.2 Reading skills across micro-level factors

In the first set of regressions, we run inverse probability weighted pooled least squares regression models by including one of the four micro factors in each of the four models: 1) household asset index quintile, 2) family members' highest educational level, 3) location (rural vs. urban), and 4) disability status. The outputs for the first stage of the selection model are presented in the Supporting Information S1 Table in S1 File. The final regression, labelled as Model 5, includes all the micro-level factor variables and control variables (Table 4).

Table 4 indicates large differences in the share of school children with satisfactory reading skills across various groups. Children from the highest quintile of the asset index outperform those from the lowest quintile by 37 percentage points (Model 1). Children in families with primary education show a 6 percentage-point advantage over those from families without any schooling, while those from families with a member who has completed junior secondary education or higher achieve a 21 percentage-point advantage (Model 2). In the full model incorporating all factors, the coefficients for wealth and education from Models 1 and 2 are reduced, likely reflecting a correlation between these factors.

Urban children outperform their rural counterparts by 23 percentage points in satisfactory reading skills before accounting for micro-level factors (Model 3) and by 9 percentage points after these factors are controlled for (Model 5).

**Table 3. Percentage of tested children with satisfactory reading skills (score > 85%) by countries, ages 10-14.**

|  | Mean (%) | Std. Err. | [95% Conf. | Interval] | Sample size | Year of survey |
|---|---|---|---|---|---|---|
| Central Africa R. | 17.8 | 0.012 | 0.155 | 0.201 | 1,080 | 2019 |
| Chad | 21.2 | 0.010 | 0.192 | 0.232 | 1,548 | 2019 |
| DRCongo | 18.9 | 0.008 | 0.175 | 0.204 | 2,730 | 2017 |
| Ghana | 47.0 | 0.009 | 0.452 | 0.488 | 2,916 | 2017 |
| Lesotho | 58.4 | 0.012 | 0.559 | 0.608 | 1,568 | 2018 |
| Madagascar | 51.2 | 0.010 | 0.492 | 0.531 | 2,477 | 2018 |
| Malawi | 49.4 | 0.007 | 0.480 | 0.508 | 4,883 | 2020 |
| The Gambia | 34.6 | 0.014 | 0.319 | 0.373 | 1,213 | 2018 |
| Togo | 37.9 | 0.012 | 0.355 | 0.403 | 1,574 | 2017 |
| Tunisia | 87.7 | 0.008 | 0.861 | 0.893 | 1,607 | 2018 |
| Zimbabwe | 56.3 | 0.011 | 0.542 | 0.585 | 2,056 | 2019 |
| **Total** | **44.7** | **0.003** | **0.441** | **0.454** | **23,652** | |

**Table 4. IPW least squares regressions on the proportion of children with satisfactory reading skills (score > 85%), by urban/rural and micro-level factors.**

| | Model1 | Model2 | Model3 | Model4 | Model5 |
|---|---|---|---|---|---|
| **Asset index (base category=Lowest quintile)** | | | | | |
| Second quintile | 0.059*** | | | | 0.044*** |
| | (0.009) | | | | (0.009) |
| Middle | 0.109*** | | | | 0.076*** |
| | (0.009) | | | | (0.010) |
| Fourth quintile | 0.209*** | | | | 0.145*** |
| | (0.010) | | | | (0.011) |
| Richest | 0.367*** | | | | 0.257*** |
| | (0.010) | | | | (0.013) |
| **Highest Educational level in the household (base category=No school)** | | | | | |
| Primary | | 0.059*** | | | 0.033*** |
| | | (0.009) | | | (0.009) |
| Junior secondary | | 0.210*** | | | 0.098*** |
| | | (0.010) | | | (0.010) |
| Senior secondary or higher | | 0.211*** | | | 0.085*** |
| | | (0.011) | | | (0.011) |
| **Location (base category: urban)** | | | -0.225*** | | -0.090*** |
| | | | (0.008) | | (0.009) |
| **Disability status (base category: non-disabled)** | | | | | |
| Vision disability | | | | 0.05 | 0.039 |
| | | | | (0.036) | (0.035) |
| Hearing disability | | | | -0.145** | -0.105* |
| | | | | (0.049) | (0.047) |
| Physical disability | | | | 0.037 | 0.073* |
| | | | | (0.035) | (0.036) |
| Intellectual disability | | | | -0.157*** | -0.150*** |
| | | | | (0.016) | (0.015) |
| Multiple disabilities | | | | -0.174*** | -0.128* |
| | | | | (0.051) | (0.050) |
| **Gender** | X | X | X | X | X |
| **Age** | X | X | X | X | X |
| **Country FE** | X | X | X | X | X |
| Sample size | 23591 | 23572 | 23591 | 23591 | 23572 |
| R2 | 0.214 | 0.176 | 0.19 | 0.153 | 0.226 |

Compared to CWOD, children with hearing disabilities (15 percentage points lower), intellectual disability (16 percentage points lower) and multiple disabilities (17 percentage points lower) exhibit lower proficiency rates (Model 4). The finding remains consistent with or without controlling for other factors (Models 4 and 5).

### 4.3 Disparities in reading skills across 11 African countries

To test hypothesis H2, we include country-specific dummy variables and interaction terms between micro-level factors and individual countries. Fig 1 presents the estimated proportion of 14-year-old children with satisfactory reading skills across various groups: rural children, children with disabilities (CWD), children from families in the lowest quintile of the asset index, and children from families without schooling. The figure also includes data on children

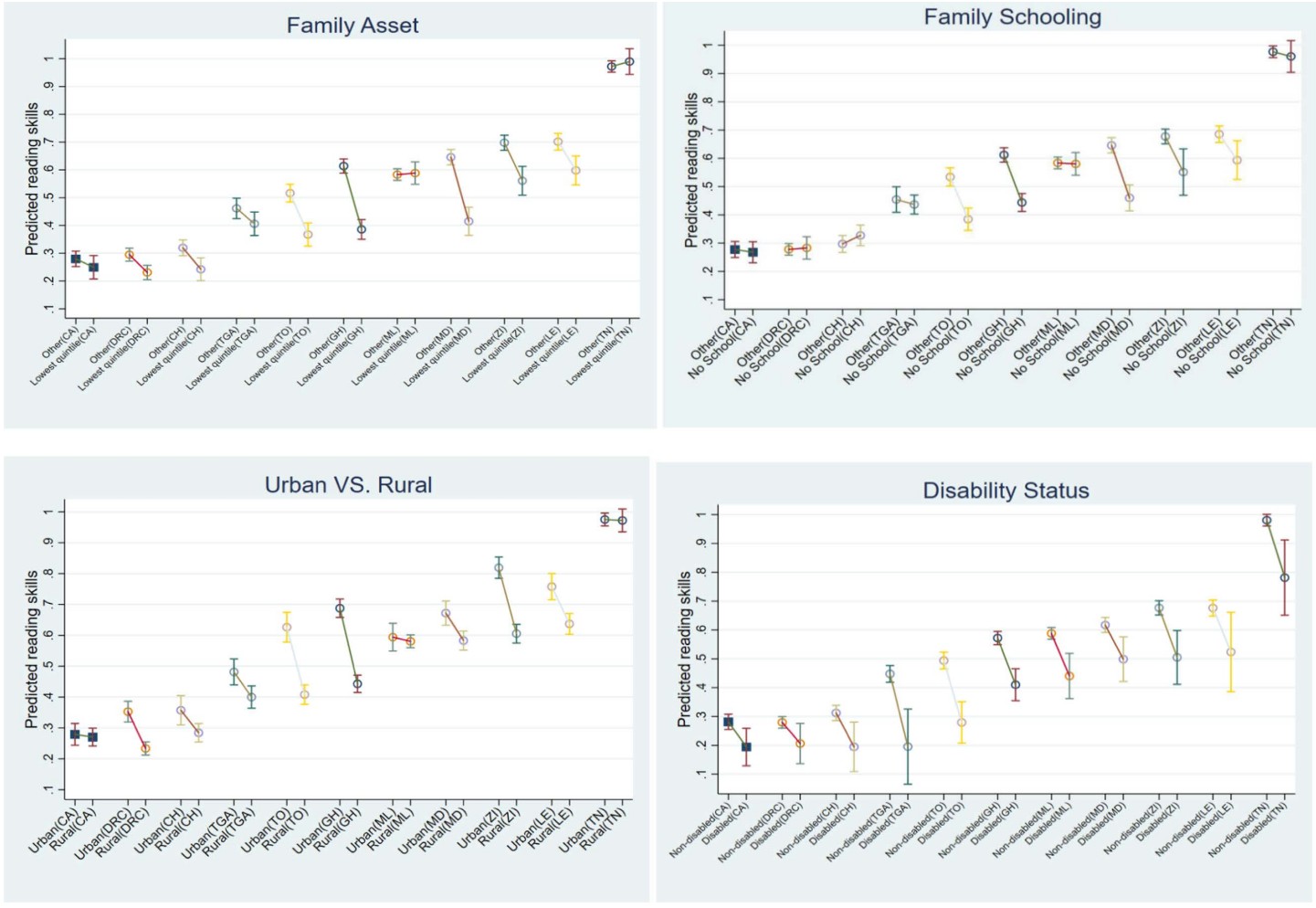

**Fig 1.  Estimated proportion of 14-year-old children with satisfactory reading skills across social groups by country, with 95% confidence intervals. Note: The predictions are calculated at the means of covariates, with separate predictions for each country. "Lowest quintile" refers to children from families in the lowest quintile of the asset index. "No school" refers to children from families without any schooling, while "Other" includes all children from families with some level of formal education. CA: Cenral Africa Republic; CH: Chad; DRC: DRCongo; GH: Ghana; LE: Lesotho; MD: Madagascar; ML: Malawi; TGA: The Gambia; TO: Togo; TN: Tunisia; ZI: Zimbabwe.**

who do not belong to these groups, offering a comparative analysis across the 11 African countries in our sample.

Disparities in reading skills between children from families in the lowest asset quintile and upper asset quintiles are significantly larger in countries with mid-level reading proficiency, such as Ghana (23 percentage points), Madagascar (23 percentage points), Togo (15 percentage points), and Zimbabwe (14 percentage points), and Lesotho (10 percentage points). In contrast, these disparities are much smaller in countries with low reading proficiencies, such as Chad (8 percentage points) and DRCongo (6 percentage points), or even no significant disparities, such as in the Central Africa Republic and The Gambia. In Tunisia, where most children have high basic reading proficiency, the differences are also insignificant. An exception is Malawi, which, despite having mid-level reading proficiency, shows no significant disparity between children from families in the lowest asset quintile and those from upper asset quintiles. Disparities in reading skills between children from families with and without

schooling have largely mirrored those from the lowest versus upper asset quintiles, with much lower disparities in countries with overall low reading proficiency.

Urban-rural disparities in reading skills are the most pronounced in Ghana (24 percentage points), Togo (22 percentage points) and Zimbabwe (21 percentage points), while they are significant but small in DRCongo (12 percentage points), Lesotho (12 percentage points), and Madagascar (8 percentage points). For other countries, the urban-rural disparities are not significant. Disparities in reading skills for children with disabilities (CWD) are significant across all 11 African countries, ranging from 7 to 22 percentage points. The largest disparity is observed in the Gambia, while countries with lower reading proficiency show smaller differences.

Despite significant disparities in reading proficiency across social groups within these countries, the cross-country differences are even more pronounced. For example, children from the lowest quintile of asset index in mid-proficiency countries such as Lesotho, Zimbabwe, Madagascar, Malawi, Ghana, Togo, and The Gambia outperform those from the top four asset quintiles in low-proficiency countries like Chad, DRCongo, and the Central African Republic.

The sample size for CWD is quite limited in several countries, resulting in a large variance in the estimated outcomes for CWD. Consequently, we further analyse the data across the three country groups defined in the first part of Result Section 4.1 (Fig 2). The results from group-level analysis are similar to those from the country-level analysis. Disparities in reading skills between children from families in the lowest and upper asset quintiles and between children from families with and without schooling are not significant in low-reading countries but are much larger in mid-reading and high-reading countries. The urban-rural disparity is especially high in the high-reading countries. However, disparities between CWD and CWOD remain consistently significant across countries with different levels of reading proficiency.

## 4.4 Disparities in reading skills related to disabilities

To test hypothesis H3, we include all micro-level indicators, as well as the interaction terms between disability status and other micro-level indicators (urban/rural residence, asset index, and family's highest educational level) in the country fixed effect model. The regression results at various cutoff points are presented in the Supporting Information S2 Table in S1 File. Sensitivity test.

Fig 3 displays the estimated proportion of 14-year-old children with satisfactory reading skills. These predictions are made with covariates set at their means for both CWD and CWOD in different social groups (urban vs. rural, high vs. low socio-economic status, more vs. less educated families). These disparities in reading skills between CWD and CWOD in schools are visually represented as lines connecting two estimated reading skill proficiency rates in various social groups. A steeper incline in the line indicates a higher disparity between CWD and CWOD, while a flatter line suggests a smaller disparity.

Fig 3 suggests that disparities in reading skills proficiency between CWD and CWOD do not vary significantly across different social groups. These disparities remain relatively constant at around 15 percentage points in various groups. The most significant disparities are observed in urban areas (19 percentage points) and among families without any schooling (21 percentage points).

Furthermore, it is noteworthy that CWD in social groups with advantaged backgrounds (urban, rich and more-educated families) have achieved similar levels of reading skill proficiency as their CWOD peers in social groups with disadvantaged backgrounds (rural, economically disadvantaged, and less-educated families).

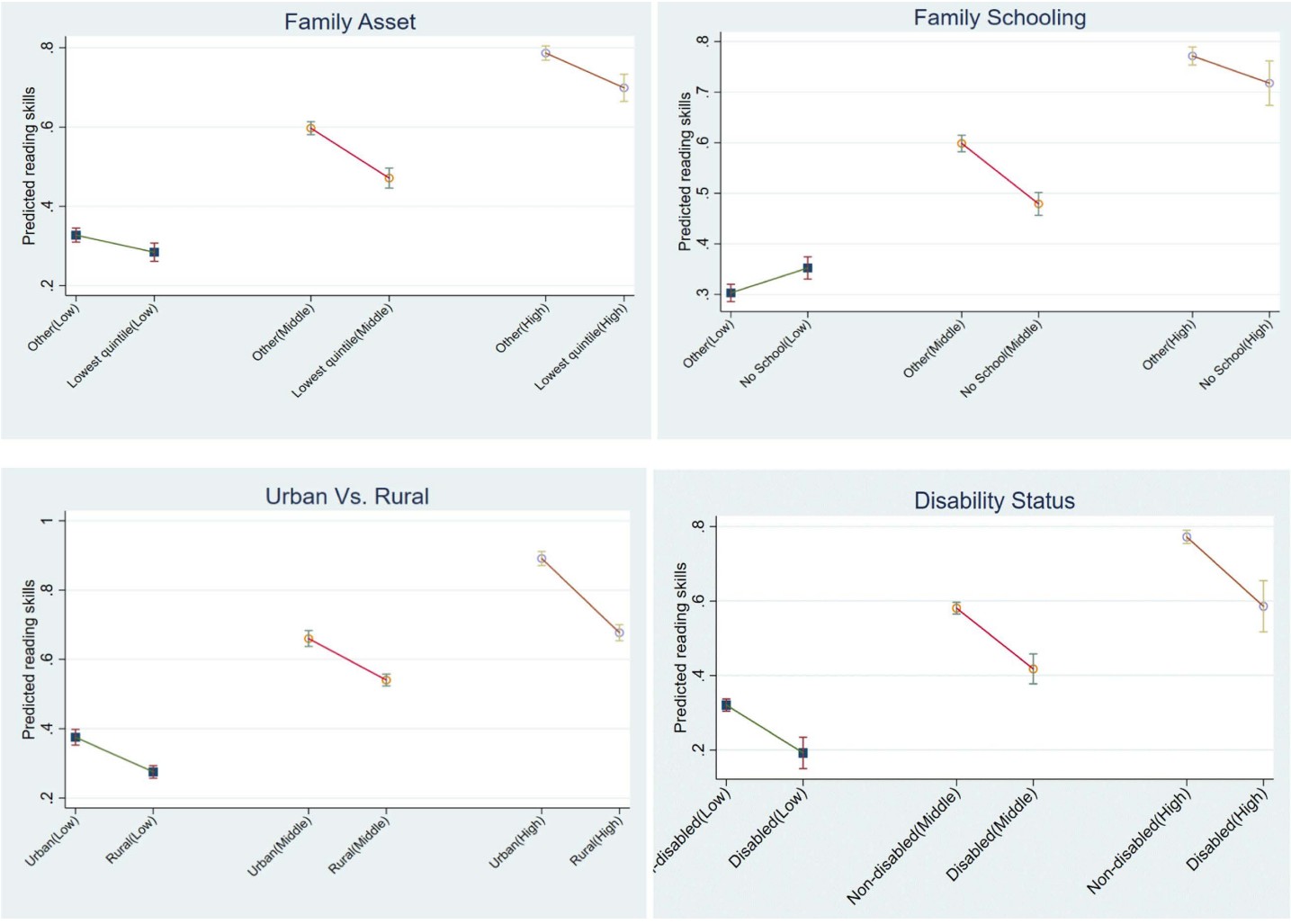

**Fig 2. Estimated proportion of 14-year-old children with satisfactory reading skills by country groups, with 95% confidence intervals.** Note: The predictions are calculated at the means of covariates, with separate predictions for each country group with low, middle or high reading skills proficiency. "Lowest quintile" refers to children from families in the lowest quintile of the asset index. "No school" refers to children from families without any schooling, while "Other" includes all children from families with some level of formal education. Low: Cenral Africa Republic, Chad, DRCongo, and The Gambia; Middle: Ghana, Madagascar, Malawi, and Togo; High: Lesotho, Tunisia, and Zimbabwe.

## 5. Discussion and study limitations

### 5.1 Discussion

In this section, we will discuss the findings related to the key hypotheses. We will also discuss important limitations of our study and provide some suggestions for future research.

Utilizing a standardized reading test, the paper reveals particularly low overall reading skills and considerable variations among school children across the 11 African countries. It is important to note that there is substantial variation in the level of school attendance across these countries, with rates ranging from 43 percent in Chad to 69 percent in Madagascar, and reaching as high as 95 percent in Lesotho, Malawi, and Tunisia. Since we expect a much lower reading skill level for children not enrolled in school, the overall reading skill level and the actual gap in learning across these countries are likely higher when differences in school

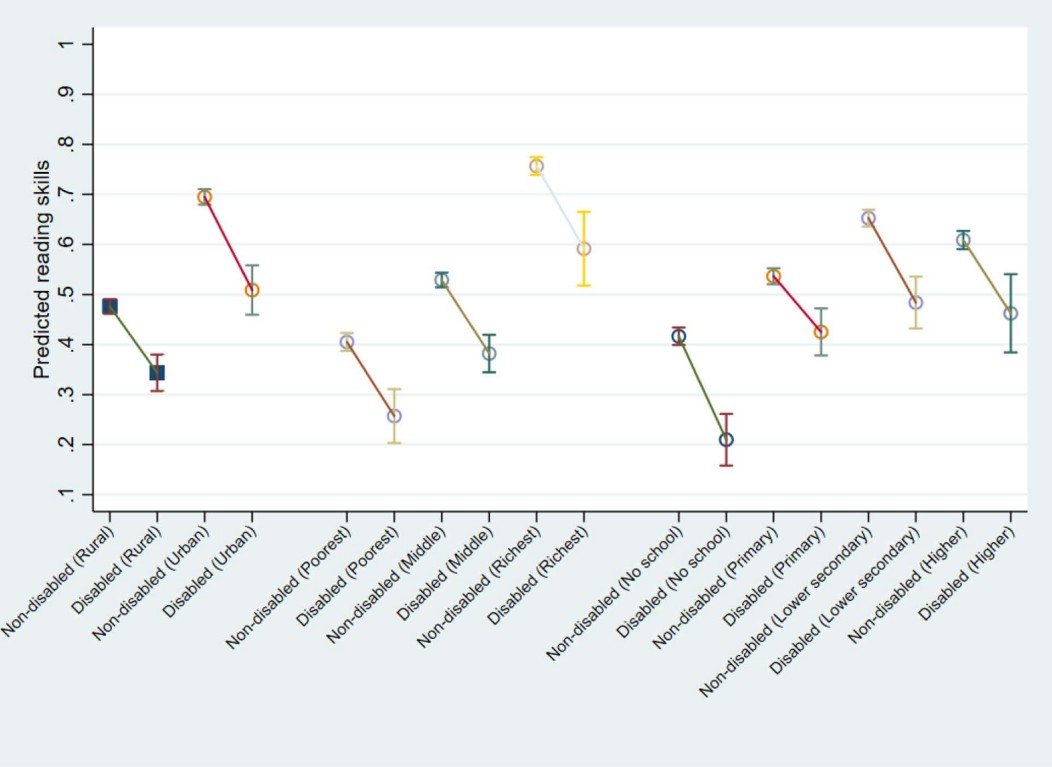

**Fig 3. Estimated proportion of 14-year-old children with satisfactory reading skills for CWD and CWOD across various social groups (Country FE), with 95% confidence intervals. Note: The predictions are calculated at the means of covariates across all countries, with separate predictions for various social groups related to rural and urban residences, family wealth index (lowest, highest and middle quintles), and the highest educational level among household members.**

attendance are considered. For instance, while the average reading skill proficiency rate is 21 percent among schoolchildren in Chad, school attendance is only 43 percent.

The first set of models supports hypothesis H1, showing that children from 1a) impoverished backgrounds, 1b) less-educated households, and 1c) rural areas exhibit significantly lower reading skills than their peers from affluent families, more educated households, or urban areas. Hypothesis H1d) is only partially supported: the percentage of school children with satisfactory reading skills is significantly lower among those with hearing, intellectual, and multiple disabilities compared to their CWOD peers [56]. However, it is important to note that children with vision or physical disabilities do not significantly lag behind, and the conclusion regarding children with hearing disabilities does not remain statistically significant when all control variables are included in the analysis.

As demonstrated by numerous studies in developed contexts [28], children from disadvantaged backgrounds tend to lag behind in reading abilities. Notably, our analysis shows that family poverty has the strongest correlation with children's reading skills. The proportion of school children in the richest quintile group who have achieved satisfactory reading skills is approximately 24–35 percentage points higher than those in the lowest quintile group.

What is particularly notable in our study is the observation that a substantial proportion of school children obtain extreme values in their reading test scores, either very low or very high scores. The concern here is primarily for school children who, at their current age, continue to achieve very low scores in basic reading tests. This underscores the substantial challenges

they may have encountered in developing proficient reading skills in the long future. Children from disadvantaged backgrounds are particularly representative.

Furthermore, our study indicates that school children with vision and physical disabilities do not exhibit significant differences in their reading skills compared to non-disabled children. It is plausible that they have managed adequately with basic reading skills. However, if more extensive reading tests were to be introduced, these children might also encounter challenges and potential difficulties in meeting advanced reading skill requirements.

Our findings support Hypothesis H2a, H2b, and H2c, indicating that disparities in reading proficiency rates across socioeconomic groups and urban-rural disparities are more pronounced in countries with higher overall reading proficiency. In countries with very low reading proficiency, such as the Central African Republic (average reading skills score of 18 percent), Chad (21per cent), and DRCongo (19 percent), disparities in reading skills across socioeconomic groups are either insignificant or much smaller compared to other countries.

The largest disparities across socioeconomic groups are observed in countries with mid-level reading proficiency, such as Ghana (47 percent), Madagascar (51 percent), Togo (38 percent), and Zimbabwe (56 percent). Urban-rural disparities are also most pronounced in countries with relatively high reading proficiency. However, in Tunisia, which boasts the highest level of socio-economic development and the highest reading proficiency (88 percent) among the 11 countries, no significant disparities in reading skills are found among children from different disadvantaged backgrounds.

Our findings do not support Hypothesis H2d, which posits that disparities in reading proficiency rates between children with and without disabilities are more pronounced in countries with higher overall reading proficiency. Meanwhile, Tunisia, the country with the highest reading proficiency (88 percent), exhibits relatively high disparities in reading skills between CWD and CWOD. However, a closer analysis reveals that the 20-percentage-point gap in Tunisia is not particularly large when viewed as a proportion of the country's overall reading proficiency level. This contrasts with the 7–12 percentage-point gaps observed in countries with significantly lower reading proficiency levels, such as the Central African Republic, Chad, and the Democratic Republic of Congo, where overall proficiency levels range from 18% to 21%. In countries with mid-level reading proficiency (35–58 percent), disparities between CWD and CWOD range from 12 to 25 percentage points, further suggesting that disability-related disparities are not significantly different across countries with different reading proficiency.

Our findings do not support Hypothesis H3 that disparities in the percentage of school children with satisfactory reading skills between CWD and CWOD would be less pronounced in households with more advantaged backgrounds. Instead, these disparities have remained relatively constant across different social groups. It is worth emphasising that these results are based on children who are currently enrolled in school. When we consider out-of-school children, recognising the overrepresentation of CWD in this group, it becomes apparent that disparities across social groups ay have been underestimated. However, as long as children are enrolled in school, a consistent gap between CWD and CWOD appears to persist.

## 5.2 Study limitations

Several limitations should be considered when interpreting the findings of this study.

First, there are some limitations associated with the reading test used in the MICS survey. Given the age range of children tested (10–14 years), the MICS reading test focuses primarily on foundational reading skills and may not assess more advanced reading skills. However, even with the basic test, the prevalence of satisfactory reading skills among children aged 10–14 in most of these countries is notably low, indicating limited reading abilities across

many African countries. Introducing a more extensive reading test could potentially reveal even larger disparities in reading proficiency.

Another limitation of the reading test is the potential for floor effects in countries with particularly low reading proficiency. The test may fail to capture important variations in the skill levels of children who do not pass it. For instance, in some countries, specific linguistic challenges—such as difficulties distinguishing between the sounds of "r" and "l"—could influence performance on the reading test, and it remains uncertain how these factors might affect the results.

Second, it is crucial to recognise that this study exclusively focuses on children currently enrolled in school. Many children not attending school and therefore not taking the reading test are disproportionately among disabled children or those from disadvantaged backgrounds. As a result, the disparities estimated in this group may have been underestimated.

Moreover, there is substantial variation in school attendance rates across the countries studied. Careful consideration is needed when analysing countries with low school enrolment. It is important to emphasise that the conclusions drawn in this paper are applicable exclusively to children enrolled in school and cannot be generalised to encompass all children in these countries.

Third, the selection of countries in this study was not guided by strict predefined criteria but was rather constrained by data availability. It is essential to interpret the estimated disparities cautiously due to the inherent arbitrariness associated with the selection of countries in this paper.

## 6. Conclusion

Our study provides new evidence on the reading proficiency of school children aged 10–14 across 11 African countries, drawing from unique nationally representative data. Through a standardized reading test, the paper uncovers notably low overall reading skills and significant disparities among school children across 11 African countries. By examining the correlations between diverse regional, familial, and individual factors, we aimed to uncover important factors that may influence school children's acquired reading skills.

A comparative analysis across 11 African countries suggests that disparities in reading skills among children from disadvantaged backgrounds are non-existent or minimal in countries with low overall reading proficiency. In contrast, these disparities are more pronounced in some countries with mid-level reading proficiency. Notably, despite having the highest overall reading proficiency, Tunisia shows no significant differences in reading skills across the social groups examined. On the other hand, given the basic nature of the reading test in this study, we can only conclude that there are no significant disparities in basic reading skills among disadvantaged children in Tunisia. However, larger disparities may emerge if more extensive reading skills are assessed.

One unique contribution of our study lies in its findings related to children with disabilities (CWD), a topic that has received relatively little attention in recent literature, likely due to data limitations. Benefiting from the large sample size from country-pooled data in the MICS standardised data, this study emphasises disparities in reading skills among children with different types of disabilities – a critical dimension often overlooked by many studies due to sample size limits.

Our study highlights a persistent gap in reading skills between CWD and CWOD across countries and various social groups, underscoring the unique challenges CWD faces. Interestingly, the differences in reading skills between CWD in poorer conditions and those in better socioeconomic conditions mirror the disparities observed among CWOD.

This paper underscores the critical role of micro-level socioeconomic factors in addressing challenges faced by vulnerable populations and enhancing reading skills for all. However, certain vulnerable groups, such as CWD, encounter unique challenges in acquiring reading skills. Although better school quality and socioeconomic conditions enhance reading skills among CWD, a significant gap between CWD and CWOD persists. Further targeted and in-depth research is essential to understand the underlying dynamics and identify tailored interventions, which extend beyond the scope of this paper.

## Supporting information

**S1 File. S1 Table Regression results from first stage of selection model for each country.** S2 Table. **Sensitivity test.** S2.1 IPW least squares regressions by micro-level factors (outcome variable cutoff at 80%). S2.2 IPW least squares regressions by micro-level factors (outcome variable cutoff at 90%). S2.3 IPW least squares regressions by micro-level factors (continuous outcome variable). S2.4 IPW least squares regressions, interaction terms between various factors and country groups (outcome variable cutoff at 85%, 80%, and 90%). S2.5 IPW least squares regressions, interaction terms between various factors and country groups (continuous outcome variable). S2.6 IPW least squares regressions, interaction terms between disability status and social factors (outcome variable cutoff at 85%, 80%, and 90%). S2.7 IPW least squares regressions, interaction terms between disability status and social factors (continuous outcome variable).
(DOCX)

## Author contributions

**Conceptualization:** Huafeng Zhang, Stein T. Holden.

**Data curation:** Huafeng Zhang.

**Formal analysis:** Huafeng Zhang.

**Methodology:** Huafeng Zhang, Stein T. Holden.

**Supervision:** Stein T. Holden.

**Writing – original draft:** Huafeng Zhang.

**Writing – review & editing:** Huafeng Zhang, Stein T. Holden.

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
