## [Decision Letter · Decision Letter 0]

14 Jan 2025

PONE-D-24-45564Disparity in School Children's Reading Skills in 11 African CountriesPLOS ONE

Dear Dr. Zhang,

Thank you for submitting your manuscript to PLOS ONE and for your patience during the review process. I have now decided to proceed based on one referee report and my reading of the manuscript. After careful consideration, we feel that it has merit but does not fully meet PLOS ONE’s publication criteria as it currently stands. Therefore, we invite you to submit a revised version of the manuscript that addresses the points raised during the review process.

The reviewer has some excellent suggestions to help improve the manuscript. There are two other issues I would like to mention: (1) In H2 and H3 you compare the percentages of school children with satisfactory basic reading skills in different groups or countries. However, your discussion of these results hints at causality. Please use caution in your interpretation of the results.  (2) The choice of variables in Equation 1 needs justification. Noting the literature examining the determinants of a child's probability of taking the reading tests or other similar outcomes in Africa or developed countries would be useful. 

We look forward to receiving your revised manuscript.

Kind regards,

Bijetri Bose

Academic Editor

PLOS ONE

4. We note that you have referenced (28. Taylor S, Yu D. The importance of socio-economic status in determining educational achievement in South Africa. Unpublished working paper (Economics) Stellenbosch: Stellenbosch University. 2009:33-47. [Cited 2024 Oct 4]. Available from: https://www.ekon.sun.ac.za/wpapers/2009/wp012009/wp-01-2009.pdf ) which has currently not yet been accepted for publication. Please remove this from your References and amend this to state in the body of your manuscript: (ie “Bewick et al. [Unpublished]”) as detailed online in our guide for authors

Reviewers' comments:

Reviewer's Responses to Questions

**Comments to the Author**

1. Is the manuscript technically sound, and do the data support the conclusions?

Reviewer #1: Yes

2. Has the statistical analysis been performed appropriately and rigorously? 

Reviewer #1: Yes

3. Have the authors made all data underlying the findings in their manuscript fully available?

Reviewer #1: Yes

4. Is the manuscript presented in an intelligible fashion and written in standard English?

Reviewer #1: Yes

5. Review Comments to the Author

Reviewer #1: 1. Summary of the research and overall impression

This manuscript provides new evidence of children’s reading skills in 11 African countries making use of Multiple Indicator Cluster Survey (MICS) data, paying particular attention to variation in reading skills by socio-economic status and urban/rural location within countries. The manuscript also considers gaps in reading skills among children with disabilities, an under-researched topic in the African context. This is an important contribution to the literature, as alarm over poor overall reading achievement levels on the continent often dwarfs evidence of variability in reading skills within countries. The fact that the manuscript makes use of nationally representative data from six countries is a particular strength.

A major limitation is that the study considers only children who are enrolled in school. This is particularly concerning when making cross-country comparisons since enrolment rates differ markedly across countries. The authors acknowledge this, and make clear that their results are only representative of children enrolled in school. Another major limitation is that floor effects might be driving the result that between-group differences are larger in countries with higher overall levels of achievement. This may be of particular concern in the African context, where sometimes 10-year-old children are unable to read a single word correctly. This limitation should be acknowledged.

Overall, the paper makes an important contribution to our understanding of differences in reading levels across different groups within countries on the African continent. My recommendation is to Accept with Major Revisions.

2. Discussion of specific areas for improvement

Major comments:

1. Introduction:

The introduction should include a discussion of the existing literature on the topic.

2. Methodology:

It is not clear why the authors use the proportion of learners achieving a specific percentage on the reading test as their outcome variable. They argue that the reason for this is the extreme values on the reading test score. Why would extreme values be a problem for estimating your coefficients of interest? I suggest sticking to a continuous variable of reading test scores.

3. Discussion:

The authors highlight the finding that many children have very low reading scores, but this is not discussed in the text where the table showing that information appears. I would suggest highlighting that early on, in the discussion of the results in Table 3.

I would also compare how learners in different groups compare across countries, as the authors compare CWD in one country to CWOD in other countries. For example, it would be nice to say “children in the lowest wealth quintile in Ghana outperform children in the top four wealth quintiles in the DRC”.

Minor comments:

1. Consider treating “children with disabilities” as a separate category, rather than lumping them together with “children from disadvantaged backgrounds”. Conceptually having a disability is very different from being from a disadvantaged background. I agree that these children are disadvantaged, but it is not their background that is disadvantaged.

2. The conclusion that “children with disabilities benefit from strong educational systems as much as children without disabilities in terms of improving their basic reading skills” does not follow from the result that children with disabilities in better-performing countries outperform children without disabilities in other countries. Rather, is suggests that strengthening education systems is a promising way of improving the reading skills of children with disabilities.

3. Use of the word “non-poor”. Operationalising poverty as being in the bottom quintile of the asset index means the proportions of “non-poor” are going to be 80% in each country by design. This might be misleading, as one would be hard pressed to argue that e.g. only 20% of people in Zimbabwe are poor.

4. Use of the word “improvements” to refer to higher country-level reading achievement. Improvements imply a change over time. I suggest rather using the phrasing “whether disadvantaged children benefit equally from attending school in countries with higher overall levels of reading achievement”.

5. Page 8: “The fundamental question revolves around whether CWD, when raised in families with a more advantageous social background (urban residence, higher income, higher education), can successfully bridge the academic performance gap compared to CWOD.” I don’t think this should be framed as the fundamental question. It is an important question, but is not set up as the main question in the rest of the paper.

6. Page 9: “Comprehensive” test should be comprehension test. The fact that the test is a oral reading fluency test followed by a comprehension test should be made clear (at the moment it is not clear before the third sentence in the second paragraph whether the story is read by an enumerator or the learner themselves (i.e. whether it is a listening comprehension test or a reading comprehension test).

7. Figures:

a. Figure 1: I strongly suggest ordering the countries by overall reading performance. This will help the reader to see that the size of the gaps between the categories are related to overall levels of reading performance in a country.

8. Consider re-structuring the conclusion. At present the discussion jumps between the findings related to disabilities in paragraph 1 to the findings related to differences between social groups in paragraph 2 back to disabilities in paragraph 3.

6. PLOS authors have the option to publish the peer review history of their article (what does this mean? ). If published, this will include your full peer review and any attached files.

**Do you want your identity to be public for this peer review?** For information about this choice, including consent withdrawal, please see our Privacy Policy .

Reviewer #1: **Yes: ** Heleen Hofmeyr

---

## [Author Response · Author response to Decision Letter 1]

29 Jan 2025

Response to Reviewer

Thank you for the constructive comments on the paper. We have carefully revised the manuscript in response to your feedback. Below, we outline the changes we made during the revision process:

Two issues raised by academic editor:

(1) In H2 and H3 you compare the percentages of school children with satisfactory basic reading skills in different groups or countries. However, your discussion of these results hints at causality. Please use caution in your interpretation of the results.

Thank you for pointing that out. We have revised the manuscript, especially the conclusion section to ensure that the results are not hinted at causality but consistently presented as comparisons across groups or countries.

(2) The choice of variables in Equation 1 needs justification. Noting the literature examining the determinants of a child's probability of taking the reading tests or other similar outcomes in Africa or developed countries would be useful.

The discussion and relevant literature regarding the selection of control variables in Equation 1 have been added to Section 3.3.

Reviewer Comments:

Reviewer #1: 1. Summary of the research and overall impression

This manuscript provides new evidence of children’s reading skills in 11 African countries making use of Multiple Indicator Cluster Survey (MICS) data, paying particular attention to variation in reading skills by socio-economic status and urban/rural location within countries. The manuscript also considers gaps in reading skills among children with disabilities, an under-researched topic in the African context. This is an important contribution to the literature, as alarm over poor overall reading achievement levels on the continent often dwarfs evidence of variability in reading skills within countries. The fact that the manuscript makes use of nationally representative data from six countries is a particular strength.

A major limitation is that the study considers only children who are enrolled in school. This is particularly concerning when making cross-country comparisons since enrolment rates differ markedly across countries. The authors acknowledge this, and make clear that their results are only representative of children enrolled in school. Another major limitation is that floor effects might be driving the result that between-group differences are larger in countries with higher overall levels of achievement. This may be of particular concern in the African context, where sometimes 10-year-old children are unable to read a single word correctly. This limitation should be acknowledged.

The limitation related to floor effects is now addressed in Section 5.2, "Study Limitations," as part of the discussion on the constraints associated with the reading test.

Overall, the paper makes an important contribution to our understanding of differences in reading levels across different groups within countries on the African continent. My recommendation is to Accept with Major Revisions.

2. Discussion of specific areas for improvement

Major comments:

1. Introduction:

The introduction should include a discussion of the existing literature on the topic.

The existing literature on the topic is limited, as most similar studies have been conducted in developed contexts. However, we identified a few relevant cross-country comparative studies in African contexts, and the discussion of these studies has been incorporated into the introduction section.

2. Methodology:

It is not clear why the authors use the proportion of learners achieving a specific percentage on the reading test as their outcome variable. They argue that the reason for this is the extreme values on the reading test score. Why would extreme values be a problem for estimating your coefficients of interest? I suggest sticking to a continuous variable of reading test scores.

Thank you for highlighting this. A discussion on the selection of the outcome variable has now been added to Section 3.1 (Data Description). Additionally, as suggested by the reviewer, we conducted the analysis using continuous reading test scores as outcome variables for all three hypotheses. The results of these analyses are presented in supplementary tables S2.3, S2.5, and S2.7 as part of the sensitivity tests, demonstrating that the choice of outcome variables does not alter the conclusions of this paper.

3. Discussion:

The authors highlight the finding that many children have very low reading scores, but this is not discussed in the text where the table showing that information appears. I would suggest highlighting that early on, in the discussion of the results in Table 3.

I would also compare how learners in different groups compare across countries, as the authors compare CWD in one country to CWOD in other countries. For example, it would be nice to say “children in the lowest wealth quintile in Ghana outperform children in the top four wealth quintiles in the DRC”.

Thank you for these valuable suggestions. The discussion of low reading scores has been integrated into the results section corresponding to Table 3. Additionally, a comparison has been made between children in the lowest wealth quintile from several mid-proficiency countries and those in higher wealth quintiles in low-proficiency countries, highlighting the significance of cross-country differences.

Minor comments:

1. Consider treating “children with disabilities” as a separate category, rather than lumping them together with “children from disadvantaged backgrounds”. Conceptually having a disability is very different from being from a disadvantaged background. I agree that these children are disadvantaged, but it is not their background that is disadvantaged.

Children with disabilities and children from other disadvantaged backgrounds are now distinctly addressed throughout the paper to ensure clarity and differentiation in the analysis and discussion.

2. The conclusion that “children with disabilities benefit from strong educational systems as much as children without disabilities in terms of improving their basic reading skills” does not follow from the result that children with disabilities in better-performing countries outperform children without disabilities in other countries. Rather, is suggests that strengthening education systems is a promising way of improving the reading skills of children with disabilities.

Thank you for the comment. The revision has been made as per the reviewer’s suggestion.

3. Use of the word “non-poor”. Operationalising poverty as being in the bottom quintile of the asset index means the proportions of “non-poor” are going to be 80% in each country by design. This might be misleading, as one would be hard pressed to argue that e.g. only 20% of people in Zimbabwe are poor.

Thank you for the suggestion. The terms "children from families in the lowest quintile of asset index" and "those from the four upper quintiles" are now used throughout the paper instead of "poor" and "non-poor" families.

4. Use of the word “improvements” to refer to higher country-level reading achievement. Improvements imply a change over time. I suggest rather using the phrasing “whether disadvantaged children benefit equally from attending school in countries with higher overall levels of reading achievement”.

Thank you for the valuable feedback. We fully agree with the reviewer’s point, and the term “improvements” has been replaced with more precise wording throughout the discussions.

5. Page 8: “The fundamental question revolves around whether CWD, when raised in families with a more advantageous social background (urban residence, higher income, higher education), can successfully bridge the academic performance gap compared to CWOD.” I don’t think this should be framed as the fundamental question. It is an important question, but is not set up as the main question in the rest of the paper.

Agree, “fundamental” is replaced in the text.

6. Page 9: “Comprehensive” test should be comprehension test. The fact that the test is a oral reading fluency test followed by a comprehension test should be made clear (at the moment it is not clear before the third sentence in the second paragraph whether the story is read by an enumerator or the learner themselves (i.e. whether it is a listening comprehension test or a reading comprehension test).

The term “comprehensive test” has been revised for clarity, and additional details have been included to explain how the reading test is conducted.

7. Figures:

a. Figure 1: I strongly suggest ordering the countries by overall reading performance. This will help the reader to see that the size of the gaps between the categories are related to overall levels of reading performance in a country.

Fig 1 has been revised to ensure that the countries are now ordered by overall reading performance, providing a clearer visual representation of the data.

8. Consider re-structuring the conclusion. At present the discussion jumps between the findings related to disabilities in paragraph 1 to the findings related to differences between social groups in paragraph 2 back to disabilities in paragraph 3.

Thank you for the suggestion. The conclusion has been restructured accordingly, with findings related to differences between social groups discussed first, followed by discussions on disabilities, to improve clarity and flow.

The figures have been uploaded to PACE tool to make it sure that they meet PLOS requirements.

We have carefully reviewed the style requirements outlined in the templates and revised the manuscript to ensure it aligns with PLOS ONE's formatting and style guidelines.

The user data file and syntax have been uploaded to figshare and here is DOI that can be used to access the data used in the analysis: DOI: 10.6084/m9.figshare.28246769

The ethics statement is now in Section 3.2 “Ethics Methods”

4. We note that you have referenced (28. Taylor S, Yu D. The importance of socio-economic status in determining educational achievement in South Africa. Unpublished working paper (Economics) Stellenbosch: Stellenbosch University. 2009:33-47. [Cited 2024 Oct 4]. Available from: https://www.ekon.sun.ac.za/wpapers/2009/wp012009/wp-01-2009.pdf) which has currently not yet been accepted for publication. Please remove this from your References and amend this to state in the body of your manuscript: (ie “Bewick et al. [Unpublished]”) as detailed online in our guide for authors

The unpublished working paper has been deleted from the reference list and revised in the body of the manuscript.

---

## [Editor Report · Decision Letter 1]

23 Feb 2025

Disparity in School Children's Reading Skills in 11 African Countries

PONE-D-24-45564R1

Dear Dr. Zhang,

Thank you for your patience. We are pleased to inform you that your manuscript has been judged scientifically suitable for publication and will be formally accepted for publication once it meets all outstanding technical requirements.

Kind regards,

Bijetri Bose

Academic Editor

PLOS ONE

---

## [Editor Report · Acceptance letter]

PONE-D-24-45564R1

PLOS ONE

Dear Dr. Zhang,

I'm pleased to inform you that your manuscript has been deemed suitable for publication in PLOS ONE. Congratulations! Your manuscript is now being handed over to our production team.

Kind regards,

on behalf of

Dr. Bijetri Bose

Academic Editor

PLOS ONE